# Experience Replay with Random Reshuffling

## Abstract

Experience replay is a key component in reinforcement learning for stabilizing learning and improving sample efficiency. Its typical implementation samples transitions with replacement from a replay buffer. In contrast, in supervised learning with a fixed dataset, it is a common practice to shuffle the dataset every epoch and consume data sequentially, which is called random reshuffling (RR). RR enjoys theoretically better convergence properties and has been shown to outperform with-replacement sampling empirically. To leverage the benefits of RR in reinforcement learning, we propose sampling methods that extend RR to experience replay, both in uniform and prioritized settings, and analyze their properties via theoretical analysis and simulations. We evaluate our sampling methods on Atari games, demonstrating their effectiveness in deep reinforcement learning.

## 1 Introduction

Reinforcement learning (RL) agents often learn from sequentially collected data, which can lead to highly correlated updates and unstable learning. Experience replay was introduced to mitigate this issue by storing past transitions and sampling them later for training updates (Lin, 1992; Mnih et al., 2015). By drawing samples from a replay memory, the agent breaks the temporal correlations in the training data, leading to more stable and sample-efficient learning (Mnih et al., 2015). Typically, these samples are drawn with replacement at random from the replay buffer. This approach treats past experiences as a reusable dataset like in supervised learning (SL).

However, there is a key difference between SL and RL in how data can be reused. In SL, one can reshuffle the entire dataset each epoch, guaranteeing that each sample is seen exactly once per epoch. In contrast, standard experience replay in RL samples experiences with replacement, meaning some experiences may be seen multiple times while others are completely missed. In the stochastic optimization literature, random reshuffling (RR)—shuffling the data each epoch—has been shown to yield faster convergence than sampling with replacement in many cases, both in theory and in practice (Bottou, 2009; 2012; HaoChen & Sra, 2019; Nagaraj et al., 2019; Rajput et al., 2020; Mishchenko et al., 2020; Gürbüzbalaban et al., 2021; Beneventano, 2023). While the assumptions of the theoretical results may not hold in RL settings in general, this insight suggests that a similar strategy in experience replay might outperform the conventional sampling method.

To explore this possibility, we propose two novel methods that integrate RR. For uniform experience replay, our method RR-C (random reshuffling with a circular buffer) applies reshuffling to the indices of a circular buffer so that each index is sampled exactly once per epoch, thereby reducing the variance of sample counts and ensuring a balanced use of past experiences. For prioritized experience replay, we introduce RR-M (random reshuffling by masking), which adapts RR to settings with dynamic transition priorities by dynamically masking transitions that have been sampled more often than expected. This strategy effectively approximates an epoch-level without-replacement sampling even as priorities continuously change.

Our contributions are summarized as follows:

- We bridge the gap between SL and RL sampling strategies by adapting RR to experience replay, addressing the unique challenges of dynamic buffer contents and changing priorities.

- For uniform experience replay, we propose RR-C, which applies RR to circular buffer indices rather than transitions themselves, ensuring balanced utilization while maintaining computational efficiency.

- For prioritized experience replay, we develop RR-M, which tracks actual versus expected sample counts and masks out priorities of oversampled transitions, providing the variance reduction benefits of RR while respecting priority-based sampling.

- We analyze our methods both theoretically and through simulations, illustrating their bias and variance properties compared to standard methods.

- We empirically demonstrate through experiments on Atari games that both RR-C and RR-M provide modest performance improvements across different deep RL algorithms: DQN, C51, and DDQN+LAP.

## 2 Preliminaries

### 2.1 Random reshuffling

In machine learning, many optimization problems involve minimizing an empirical loss $F(w) = \frac{1}{N} \sum_{i=1}^{N} f_i(w)$, where each $f_i(w)$ represents the loss on a data sample. Stochastic gradient descent (SGD) is a prevalent method for such tasks, especially when $N$ is large. At each iteration $t$, SGD randomly selects an index $i_t \in \{1, \ldots, N\}$ and updates the parameter $w$ as $w_{t+1} = w_t - \eta_t \nabla f_{i_t}(w_t)$, with $\eta_t$ being the learning rate. Sampling $i_t$ with replacement from $\{1, \ldots, N\}$ provides an unbiased estimate of the full gradient $\nabla F(w_t)$, supporting convergence guarantees under appropriate conditions (Robbins & Monro, 1951).

Random reshuffling (RR) is an alternative way of sampling $i_t$ that shuffles the order of the dataset $\{1, \ldots, N\}$ every epoch and sequentially processes each sample exactly once per epoch. This approach ensures uniform data coverage within each cycle, potentially reducing variance in updates. Not only is it widely used in practice, but RR has also been shown to outperform with-replacement sampling theoretically under certain conditions; for example, achieving a faster convergence rate of $O(1/K^2)$ compared to the $O(1/K)$ rate of with-replacement sampling SGD for smooth and strongly convex loss functions, given a sufficiently large number of epochs $K$ (Nagaraj et al., 2019).

### 2.2 Experience replay

Experience replay is a technique for RL that stores past transitions $(s_t, a_t, r_t, s_{t+1})$ in a replay buffer and samples a minibatch from the buffer to update the parameters of a neural network (Lin, 1992; Mnih et al., 2015). Typically, the replay buffer is a FIFO (first-in first-out) queue with a fixed capacity and implemented as a circular buffer, where new transitions overwrite old transitions when the buffer is full. If the probability of sampling a transition is uniform across the buffer, we call it uniform experience replay.

Prioritized experience replay (Schaul et al., 2016) is a popular variant of experience replay that assigns a priority to each transition and samples transitions with a probability proportional to their priority. How to compute the priority of a transition is a key design choice in prioritized experience replay, and several methods have been proposed in the literature (Fujimoto et al., 2020; Oh et al., 2022; Saglam et al., 2023; Sujit et al., 2023). The probability of sampling a transition is defined as $P(i) = p_i / \sum_k p_k$, where $p_i$ is the priority of transition $i$. The priority is updated after a transition is sampled, which makes it non-stationary. We can consider uniform experience replay as a special case of prioritized experience replay where the priority of each transition is fixed to a global constant. Typical implementations use a sum tree data structure to efficiently sample transitions according to priorities (Schaul et al., 2016).

# 3 Experience replay with random reshuffling

## 3.1 Sampling methods for experience replay

Given the probabilities of transitions, flexibility remains in how minibatches are sampled according to these probabilities. We summarize the sampling methods of experience replay in popular RL code bases in table 3. The most common method for both uniform and prioritized experience replay is to sample transitions with replacement, where the same transition can be sampled multiple times in a minibatch. The sample code for uniform experience replay is shown as `sample_with_replacement` in figure 1. Other approaches include within-minibatch without-replacement sampling (shown as `sample_without_replacement` in figure 1) and stratified sampling that segments the range of query values into strata and samples from each stratum, both of which prevent sampling the same transition multiple times in a minibatch.

These sampling methods still exhibit variance in the number of times each transition is sampled throughout the training process. It is possible that some transitions are never sampled while other transitions are sampled multiple times even in the case of uniform experience replay, where each transition is considered equally important. Intuitively, this variance is unnecessary and can be mitigated by a better sampling method like RR. The variance reduction could lead to more stable learning and better sample efficiency. Why is RR not used in experience replay?

While RR has been shown to be superior in SL, it is not directly applicable to experience replay in RL[1]. Both the size and the content of the replay buffer are dynamically changing, which makes it difficult to apply RR directly. Prioritized experience replay introduces additional complexity, where the priority of a transition is not fixed and changes every time it is sampled. We tackle these challenges by proposing two sampling methods that extend RR to experience replay, both in uniform and prioritized settings, which we describe in the following sections.

We note that transitions are collected sequentially and thus are Markovian in typical RL settings, not i.i.d. as in SL, which can also affect the effectiveness of RR in RL. Still, we hypothesize that the variance reduction of RR can help in RL as well, which is supported by our empirical results.

## 3.2 Uniform experience replay with random reshuffling

Our aim is to extend RR to experience replay in a way that satisfies the following properties:

- **Equivalence to RR**: If the transitions in the buffer are fixed, the sampling method should work just like RR in SL.

- **Variance reduction**: Even if the transitions in the buffer are changing, the sampling method should reduce the variance of how many times each transition is sampled while not introducing significant bias.

We propose a simple extension of RR that satisfies these properties: applying RR to the indices of a circular buffer, rather than to the transitions themselves, assuming that the replay buffer is implemented as a circular buffer. We call this method random reshuffling with a circular buffer (RR-C).

The sample code of RR-C is shown as `sample_rrc` of figure 1. Specifically, we create a shuffled list of indices $[0, 1, \ldots, C - 1]$ (`rr_buffer` in the figure), where $C$ is the capacity of the circular buffer, and sequentially consume the list when we sample a minibatch during an epoch. When we reach the end of the list, we create a new shuffled list of indices and start a new epoch. It is easy to see that each index is sampled exactly once in an epoch, which is the key property of RR. Although old transitions are overwritten by new transitions in a circular buffer as RL training proceeds, if a transition stays in the buffer for at least $n$ whole epochs, we can guarantee that it is sampled at least $n$ times. If a sampled index has not been assigned to a transition yet, we simply skip it and continue with the next.

---

[1]Offline RL algorithms with a fixed dataset and online RL algorithms that repeat minibatch updates over the latest batch of transitions like PPO (Schulman et al., 2017) can be exceptions if we define experience replay so that it includes these cases.

This method is easy to implement and has no hyperparameters to be tuned. It is also computationally efficient because we only need to make a shuffled list of indices at the beginning of each epoch.

```python
import numpy

def sample_with_replacement(transitions: list, batch_size: int) -> list:
    """Sample with replacement."""
    indices = numpy.random.randint(0, len(transitions), batch_size)
    return [transitions[i] for i in indices]

def sample_without_replacement(transitions: list, batch_size: int) -> list:
    """Sample without replacement."""
    indices = numpy.random.choice(len(transitions), batch_size, replace=False)
    return [transitions[i] for i in indices]

def sample_rrc(transitions: list, batch_size: int, max_size: int, rr_buffer: list) -> list:
    """Sample with RR-C.

    max_size: the capacity of the replay buffer
    rr_buffer: a list of shuffled indices
    """
    indices = []
    while len(indices) < batch_size:
        if len(rr_buffer) == 0:
            rr_indices = numpy.arange(0, max_size)
            numpy.random.shuffle(rr_indices)
            rr_buffer[:] = list(rr_indices)
        i = rr_buffer.pop()
        if i < len(transitions):
            indices.append(i)
    return [transitions[i] for i in indices]
```

Figure 1: Python code examples of sampling methods for uniform experience replay. This simplified code is for illustrative purposes and is not identical to actual implementations.

Our theoretical analysis shows that RR-C is not strictly unbiased in the early stage of training, but it becomes unbiased and lower variance in later timesteps. If we denote by $X_{i,k}^{\mathrm{UER}}$ and $X_{i,k}^{\mathrm{RR\text{-}C}}$ the number of times transition $i$ (i.e., the transition added to the buffer at timestep $t = i$) is sampled up to timestep $k$ in uniform experience replay with with-replacement sampling (UER) and RR-C, respectively, we have the following results.

**Theorem 1.** $\mathbb{E}\left[X_{i,k}^{RR\text{-}C}\right] = \mathbb{E}\left[X_{i,k}^{UER}\right]$ *holds for all $(i, k)$ for sufficiently large $i$.*

**Theorem 2.** $\mathrm{Var}\left[X_{i,k}^{RR\text{-}C}\right] \leq \mathrm{Var}\left[X_{i,k}^{UER}\right]$ *holds for all $(i, k)$ such that $i \leq k$ with sufficiently large $i$. The equality holds only under a few specific conditions.*

See appendix B for details of the analysis and proofs.

### 3.3 Prioritized experience replay with random reshuffling

Similarly to the uniform case, we aim to extend RR to prioritized experience replay in a way that satisfies the following properties:

- **Reducibility to RR**: If the transitions in the buffer are fixed and their probabilities are all the same, it should work just like RR in SL.

- **Epoch equivalence to RR with repeated data**: If the priorities of transitions are all rational numbers, we can consider an epoch where each transition appears exactly how many times it should be sampled in expectation. For example, given three transitions whose priorities are fixed to [1, 0.5, 2], for an epoch of length 7, the sampling method should sample the first transition twice, the second once, and the third four times.

- **Variance reduction**: Similarly to the uniform case, even if the transitions in the buffer are changing and their priorities are non-stationary, the sampling method should reduce the variance of how many times each transition is sampled while not introducing significant bias.

For a simple case, it seems possible to make a list of shuffled indices and consume it sequentially as in RR-C to satisfy the equivalence to RR with repeated data, e.g., by shuffling the list $[0, 0, 1, 2, 2, 2, 2]$ for the example above. However, this quickly becomes impractical for large buffer sizes. Also, it is not clear how to reflect the updated priorities of transitions in the shuffled list.

We propose a method that satisfies the above properties by masking the priorities of transitions that have been oversampled relative to expectation, which we call RR by masking (RR-M). The sample code of this method is shown as `sample_rrm` in figure 2. It keeps actual and expected sample counts of transitions and updates them every time a minibatch is sampled. When a transition is deemed oversampled, it is masked by setting its priority to a tiny value[2] until its expected count catches up. When old transitions are overwritten by new transitions, we reset the actual and expected counts of the overwritten transitions to zero and scale the expected counts so their sum matches the sum of actual counts. When we sample a minibatch, we do within-minibatch without-replacement sampling according to the masked priorities. This can be achieved by querying the sum tree in a batch manner, removing the duplicated indices if any, temporarily masking the priorities of the sampled transitions, resample only the amount that is less than the desired minibatch size, and repeating the process until the minibatch is filled. We include example trajectories of the behavior of RR-M in table 4.

This method is also easy to implement and has no hyperparameters to be tuned, but it is relatively computationally expensive because every time we sample a minibatch, we need to update the counts of transitions, which takes $O(n)$ time, where $n$ is the number of transitions in the replay buffer. In practice, it is easy to parallelize the computation of the counts and oversampled indices, e.g., with GPUs. We discuss the computational cost in more detail in appendix D.

```python
import numpy

def sample_rrm(transitions: list, sumtree: Any, batch_size: int, actual_counts: numpy.
    ndarray, expected_counts: numpy.ndarray) -> list:
    """Sample with RR-M.

    sumtree: a sum tree data structure priorities and masks
    actual_counts: how many times each transition has been sampled
    expected_counts: how many times each transition should have been sampled in expectation
    """
    # Check if each transition is oversampled (by elementwise comparison).
    oversampled = numpy.greater(actual_counts, expected_counts)
    # Mask oversampled transitions and unmask others.
    sumtree.update_mask(oversampled)
    # Sample without replacement according to masked priorities.
    indices = sumtree.sample(batch_size)
    # Update counts.
    for i in indices:
        actual_counts[i] += 1
    expected_counts += sumtree.probabilities * batch_size
    return [transitions[i] for i in indices]
```

Figure 2: A Python code example of our sampling method for prioritized experience replay, RR-M. This simplified code is for illustrative purposes and is not identical to actual implementations.

Our theoretical analysis shows that RR-M is biased even for later timesteps. Yet, under a certain simplified setting, it can be proved that its relative error to the expected sample counts vanishes and achieves lower variance asymptotically. See appendix B for details of the analysis and proofs.

---

[2]In our experiments, we multiply the original priority by 1e-8, not 0, so that the algorithm would work if all the transitions were deemed oversampled due to numerical errors.

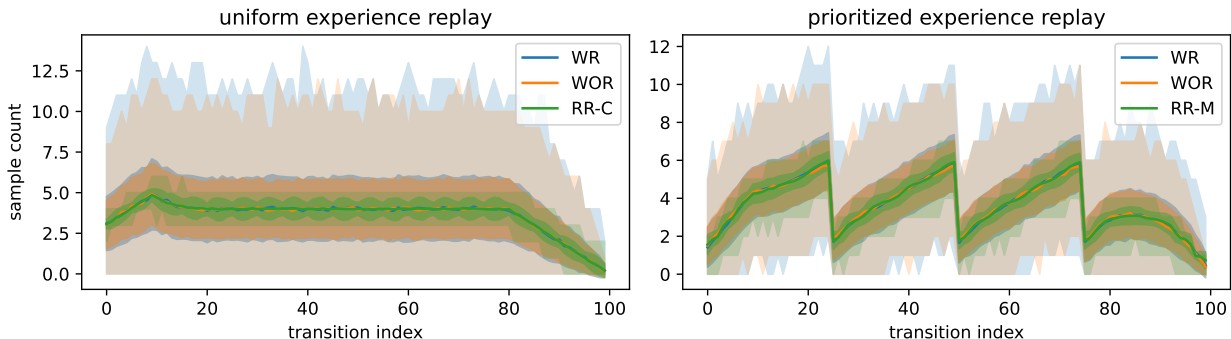

Figure 3: Distributions of sample counts in experience replay simulations. We ran 100-timestep simulations with different random seeds for each configuration. For each transition, we visualize how many times it is sampled during a simulation. Solid lines represent mean sample counts, thick shaded areas represent mean±stdev, and light shaded areas represent minimum to maximum over 1000 simulations. WR: with-replacement sampling, WOR: without-replacement sampling, RR-C: RR with a circular buffer, RR-M: RR by masking.

# 4 Experiments

## 4.1 Experience replay simulations

We first conduct simple numerical simulations of experience replay to illustrate the benefits of our RR-based sampling methods, RR-C and RR-M. In a 100-timestep simulation, an agent sequentially observes transitions $T_t$ indexed by timesteps $t = 0, \ldots, 99$. At timestep $t$, the agent stores transition $T_t$ in a replay buffer of capacity 20. If the buffer contains at least 10 transitions, the agent samples a minibatch of size 4 from the buffer. We record the indices of sampled transitions and visualize the distribution of their sample counts for each transition across 1000 simulations using different random seeds.

The left side of figure 3 shows the distributions of sample counts in uniform experience replay simulations. When we compare with-replacement sampling and without-replacement sampling, we observe that without-replacement sampling exhibits slightly lower-variance sample counts. However, the variance is still high; occasionally, some transitions are sampled more than 10 times while others may not be sampled at all. RR-C further reduces the variance of sample counts without noticeable bias; each transition is sampled at most 6 times.

Similarly, the right side of figure 3 shows the distributions of sample counts in prioritized experience replay simulations. We simulate non-uniform and dynamic priorities; the priority of each transition is set to $p_t = (t \bmod 25) + 5$ when added to the buffer and decays by a factor of 0.8 every time it is sampled, where these values are chosen arbitrarily for illustration. Again, we observe that RR-M realizes lower-variance sample counts without significant bias. We include additional simulation results with different set of parameters in section E.1.

From these simulations, we observe that both our RR-based sampling methods can reduce the variance of sample counts in experience replay without introducing significant bias, which complements our theoretical analysis.

## 4.2 Deep RL with uniform experience replay

To evaluate the effectiveness of experience replay with RR in deep RL settings, we conduct experiments on the Arcade Learning Environments (Bellemare et al., 2013). To save computational costs, we use the 10 Atari games referred to as Atari-10 (Aitchison et al., 2023), which is chosen to be small but representative.

We run the DQN (Mnih et al., 2015) and C51 (Bellemare et al., 2017) algorithms and compare the performance of different sampling methods. We base our implementation on the CleanRL library (Huang et al., 2022) and keep the hyperparameters the same as the default values. We run each experiment with 10 random seeds and report the final performance, measured by the mean score over the last 100 episodes. Learning curves are shown in section E.4 in Appendix.

While our main focus is to evaluate and compare per-game performance of sampling methods for better understanding of the effectiveness of the methods, the Atari-10 subset is accompanied by the regression coefficients that allow us to extrapolate the scores to the full 57-game performance (Aitchison et al., 2023), which we include in section E.5 in Appendix.

Table 1: Final performance of C51 and DQN with with-replacement sampling (WR) and RR-C. Final performance is measured by the mean episodic returns of the last 100 training episodes and reported as the mean over 10 random seeds with higher values bold. The p-values are computed by Welch's t-test. *: $p < 0.05$, **: $p < 0.01$. Tables with standard deviations are included in section E.3 in Appendix.

|  | C51 | | | DQN | | |
|  | WR | RR-C | p-value | WR | RR-C | p-value |
|---|---|---|---|---|---|---|
| Amidar | 347.63 | **420.61** | 1.1e-02* | 319.04 | **341.44** | 3.2e-01 |
| Bowling | **39.45** | 33.90 | 2.5e-01 | 42.17 | **43.97** | 6.2e-01 |
| Frostbite | 3568.36 | **4073.23** | 3.9e-02* | **2351.80** | 1897.50 | 2.4e-01 |
| KungFuMaster | 21091.90 | **21407.00** | 7.7e-01 | 3674.70 | **4291.40** | 8.3e-01 |
| Riverraid | 11984.34 | **12776.65** | 6.0e-02 | 8346.39 | **8513.25** | 6.1e-01 |
| BattleZone | 22483.00 | **23776.00** | 1.9e-02* | 22651.00 | **24546.00** | 3.9e-03** |
| DoubleDunk | **-15.04** | -16.42 | 2.2e-02* | -17.86 | **-14.69** | 2.1e-01 |
| NameThisGame | 8287.14 | **8844.65** | 1.9e-03** | 6064.81 | **6919.35** | 1.3e-04** |
| Phoenix | 11920.93 | **13749.39** | 7.7e-03** | 9059.42 | **10194.60** | 6.8e-02 |
| Qbert | 15685.83 | **16352.80** | 6.4e-02 | 13191.77 | **13859.60** | 1.1e-01 |

Table 1 shows the final performance of C51 and DQN with different sampling methods. Both C51 and DQN with RR-C outperform their with-replacement sampling counterparts in the majority of games, which demonstrates the effectiveness of RR in experience replay.

**Is within-minibatch without-replacement sampling sufficient?** We also evaluate the performance of C51 with within-minibatch without-replacement sampling and observe that it is no better than with-replacement sampling, which suggests that the benefits of RR come from the buffer-level without-replacement sampling. This is not surprising because sampling with replacement in this setting rarely samples duplicate transitions in the same minibatch; the capacity of the buffer is set to 1 million, which is much larger than the minibatch size of 32. The final performance is shown in figure 7.

### 4.3 Deep RL with prioritized experience replay

To compare sampling methods for prioritized experience replay, we run the double DQN algorithm (van Hasselt et al., 2016) with loss-adjusted experience replay (DDQN+LAP) (Fujimoto et al., 2020) and compare the performance against with-replacement sampling. We use their official code base and keep the hyperparameters[3] the same as the default values except for the number of training timesteps, which is set to 10 million steps across all experiments due to the limitation of computational resources. As in the uniform experience replay experiments, we run each experiment with 10 random seeds and report the final performance which is measured by the mean score over the last 100 episodes.

Table 2 shows the final performance of DDQN+LAP with different sampling methods. The left side of the table compares with-replacement sampling and RR-M. Although the differences are less pronounced

---

[3]Notably, we set action repeat to 0.25 for DDQN+LAP (following the original paper) and to 0 for C51 and DQN.

compared to uniform experience replay experiments, RR-M outperforms with-replacement sampling in 8 out of 10 games, which suggests that RR can also be beneficial in prioritized experience replay.

**Is stratified sampling sufficient?** Stratified sampling is sometimes used in prioritized experience replay, querying a different segment of the sum tree for each transition in a minibatch. Since priorities are stored in a sum tree in chronological order, stratified sampling ensures diversity in age of transitions in a minibatch, which could have a different benefit for learning. The right side of table 2 compares stratified sampling and RR-M+ST, where stratified sampling is used instead of within-minibatch without-replacement sampling. We observe that RR-M+ST outperforms stratified sampling in 8 out of 10 games, which suggests that RR can be beneficial even when stratified sampling is used.

Table 2: Final performance of DDQN+LAP with with-replacement sampling (WR), RR-M, stratified sampling (ST), and RR-M+ST (ST applied to RR-M). Format follows table 1. Tables with standard deviations are included in section E.3 in Appendix.

| | DDQN+LAP | | | | | |
| | WR | RR-M | p-value | ST | RR-M+ST | p-value |
|---|---|---|---|---|---|---|
| Amidar | **196.94** | 195.35 | 9.0e-01 | 181.64 | **206.07** | 2.5e-02* |
| Bowling | 28.85 | **29.87** | 6.3e-01 | **34.86** | 27.57 | 3.8e-02* |
| Frostbite | 1602.00 | **1761.34** | 7.3e-02 | 1672.56 | **1684.80** | 9.0e-01 |
| KungFuMaster | 16628.90 | **17580.60** | 1.4e-01 | 16859.30 | **17337.90** | 4.7e-01 |
| Riverraid | 7609.97 | **7756.11** | 1.6e-01 | 7474.01 | **7810.06** | 2.0e-02* |
| BattleZone | **22897.00** | 20813.00 | 1.8e-01 | 22099.00 | **22584.00** | 5.4e-01 |
| DoubleDunk | -17.49 | **-17.47** | 9.5e-01 | -17.34 | **-17.18** | 8.2e-01 |
| NameThisGame | 2589.54 | **2805.85** | 4.9e-03** | 2635.89 | **2759.89** | 1.0e-01 |
| Phoenix | 4180.98 | **4342.48** | 1.3e-01 | 4214.82 | **4399.41** | 1.3e-01 |
| Qbert | 4128.27 | **4266.98** | 6.1e-01 | **4109.27** | 4017.55 | 6.6e-01 |

## 5 Related work

Experience replay is a well-established technique in off-policy RL, originally proposed by Lin (1992) and later popularized in deep Q-networks by Mnih et al. (2015). This mechanism has become a cornerstone of many deep RL algorithms, including DQN and its variants (van Hasselt et al., 2016; Bellemare et al., 2017). Prioritized experience replay (Schaul et al., 2016) biases sampling towards important transitions (e.g. those with high temporal-difference error), which can accelerate learning by replaying valuable experiences more frequently, followed by extensions that differ in how priorities are computed (Fujimoto et al., 2020; Oh et al., 2022; Saglam et al., 2023; Sujit et al., 2023; Yenicesu et al., 2024). Our proposed sampling methods can be used as a drop-in replacement to with-replacement sampling for both uniform and prioritized experience replay. They can be combined with extensions that are orthogonal to how transitions are sampled such as hindsight experience replay (Andrychowicz et al., 2017), which relabels goals for multi-goal RL tasks to improve sample efficiency.

Several studies have investigated sampling methods for experience replay. Panahi et al. (2024) reported within-minibatch without-replacement sampling has a minor benefit over sampling with replacement in prioritized experience replay in tabular settings with smaller buffer sizes. Closely related to our work are combined experience replay (CER) (Zhang & Sutton, 2017) and corrected uniform experience replay (CUER) (Yenicesu et al., 2024); CER ensures the most recent transitions are always included in a minibatch to mitigate the negative effects of a large replay buffer, while CUER, a variant of prioritized experience replay, prioritizes transitions that have been sampled less often by reducing the priority every time a transition is sampled. Similarly to our work, both CER and CUER could reduce the variance in the number of times each transition is sampled, as CER ensures that each transition is sampled at least once and CUER reduces the priority of transitions that have been sampled more often than others. However, our RR-based sampling methods aim to eliminate such unnecessary variance by either enforcing that each transition is sampled exactly once per

epoch (RR-C) or detecting and excluding the oversampled transitions (RR-M). While CER and CUER favor recent transitions by design, our RR-based sampling methods do not have such a bias and can be applied to both uniform and prioritized experience replay.

## 6 Conclusion

In this work, we introduced two sampling methods that extend random reshuffling to uniform and prioritized experience replay, respectively. Our theoretical analysis and simple simulations suggest that our methods may reduce the variance in the number of times each transition is sampled without introducing significant bias. Our empirical results on Atari games demonstrate that introducing RR to experience replay yields modest improvements across different RL algorithms. Importantly, our RR-based sampling methods are straightforward to implement and can safely replace with-replacement sampling in existing RL code bases, making them practical for RL practitioners.

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

# A  Sampling methods of experience replay in popular RL code bases

We summarize the sampling methods of experience replay in popular RL code bases in table 3.

Table 3: Sampling methods of experience replay in popular RL code bases. WR: with-replacement sampling, WOR: (within-minibatch) without-replacement sampling, ST: stratified sampling. Code bases are sorted in alphabetical order. URLs were checked around February 23, 2025.

| Code base | Uniform ER | Prioritized ER |
|---|---|---|
| CleanRL (Huang et al., 2022) | WR[a] | N/A |
| Dopamine (Castro et al., 2018) | WR[b] | ST[c] |
| DQN 3.0 (Mnih et al., 2015) | WR[d] | N/A |
| DQN Zoo (Quan & Ostrovski, 2020) | WR[e] | WR[f] |
| OpenAI Baselines (Dhariwal et al., 2017) | WR[g] | ST[h] |
| PFRL (Fujita et al., 2021) | WOR[i] | WOR[j] |
| Ray RLlib (Liang et al., 2018) | WR[k] | WR[l] |
| Stable-Baselines3 (Raffin et al., 2021) | WR[m] | N/A |
| Tianshou (Weng et al., 2022) | WR[n] | WR[o] |
| TorchRL (Bou et al., 2023) | WR[p q] | WR[r] |

[a]CleanRL depends on Stable-Baselines3's `ReplayBuffer`

[b]https://github.com/google/dopamine/blob/bec5f4e108b0572e58fc1af73136e978237c8463/dopamine/tf/replay_memory/circular_replay_buffer.py#L552

[c]https://github.com/google/dopamine/blob/bec5f4e108b0572e58fc1af73136e978237c8463/dopamine/tf/replay_memory/prioritized_replay_buffer.py#L160

[d]https://github.com/google-deepmind/dqn/blob/9d9b1d13a2b491d6ebd4d046740c511c662bbe0f/dqn/TransitionTable.lua#L124

[e]https://github.com/google-deepmind/dqn_zoo/blob/45061f4bbbcfa87d11bbba3cfc2305a650a41c26/dqn_zoo/replay.py#L158

[f]https://github.com/google-deepmind/dqn_zoo/blob/45061f4bbbcfa87d11bbba3cfc2305a650a41c26/dqn_zoo/replay.py#L559

[g]https://github.com/openai/baselines/blob/ea25b9e8b234e6ee1bca43083f8f3cf974143998/baselines/deepq/replay_buffer.py#L67

[h]https://github.com/openai/baselines/blob/ea25b9e8b234e6ee1bca43083f8f3cf974143998/baselines/deepq/replay_buffer.py#L112

[i]https://github.com/pfnet/pfrl/blob/c8cb3328899509be9ca68ca9a6cd2bcfce95725c/pfrl/collections/random_access_queue.py#L101

[j]https://github.com/pfnet/pfrl/blob/c8cb3328899509be9ca68ca9a6cd2bcfce95725c/pfrl/collections/prioritized.py#L303

[k]https://github.com/ray-project/ray/blob/db419295a60657fc2be7e0e8053cea9ee8b82fbd/rllib/utils/replay_buffers/replay_buffer.py#L314

[l]https://github.com/ray-project/ray/blob/db419295a60657fc2be7e0e8053cea9ee8b82fbd/rllib/utils/replay_buffers/prioritized_replay_buffer.py#L88

[m]https://github.com/DLR-RM/stable-baselines3/blob/fa21bce04ee625c67f6ea2a7678bf46c39cd226c/stable_baselines3/common/buffers.py#L114

[n]https://github.com/thu-ml/tianshou/blob/c4ae7cd924cc76b3c3f456008ce4df80d3d8c746/tianshou/data/buffer/base.py#L502

[o]https://github.com/thu-ml/tianshou/blob/0a79016cf6f6c7f44aa5d6a1af36a96247bb1e78/tianshou/data/buffer/prio.py#L65

[p]https://github.com/pytorch/rl/blob/21c4d87c7ba5592a731757411468d080b7173b5f/torchrl/data/replay_buffers/storages.py#L159

[q]Additionally, TorchRL also supports `SamplerWithoutReplacement` sampler, which resembles our RR-C. However, it resets its list of shuffled indices every time the storage is expanded, thus making it unsuitable for a replay buffer of a growing size, which is typical in off-policy online RL training. https://github.com/pytorch/rl/blob/21c4d87c7ba5592a731757411468d080b7173b5f/torchrl/data/replay_buffers/samplers.py#L162

[r]https://github.com/pytorch/rl/blob/21c4d87c7ba5592a731757411468d080b7173b5f/torchrl/data/replay_buffers/samplers.py#L497

# B  Theoretical results

In this section, we give theoretical results with proofs on our proposed sampling methods, RR-C and RR-M.

## B.1  Notations

We denote by $C$ the capacity of the replay buffer and by $B$ the minibatch size. For every timestep $t = 0, 1, \ldots$ we add a new transition $T_t$ to the buffer, and, if the buffer contains at least $R$ transitions, sample a minibatch of size $B$ from the buffer. $B \leq R \leq C$ always holds. For prioritized settings, let $p_i$ be the priority of transition $T_i$ in the buffer.

The number of times transition $T_i$ is sampled up to timestep $t = k$ is a random variable, denoted $X_{i,k}$, where randomness comes from the sampling procedure. Since $T_i$ is added to the replay buffer at timestep $t = i$ and dropped at timestep $t = i + C$, $X_{i,k} = 0$ for all $k < i$ and $X_{i,k} = X_{i,i+C-1}$ for all $k \geq i + C - 1$. Let $t_{\text{start}}(i) := \max\{i, \ R - 1\}$ and $t_{\text{end}}(i, k) := \min\{k, \ i + C - 1\}$. The number of timesteps on which minibatches are actually drawn while $T_i$ is present is $N_{i,k} := \max\{0, \ t_{\text{end}}(i, k) - t_{\text{start}}(i) + 1\}$, and we define $S_{i,k} := B \cdot N_{i,k}$. In particular, when $i \geq C$ we have $t_{\text{start}}(i) = i$ and $t_{\text{end}}(i, k) = \min\{k, i + C - 1\}$, so $S_{i,k} = B \cdot \min(k - i + 1, C)$, which is the expression we will use in the proofs below.

We represent which sampling method is used by a superscript of a random variable, so $X_{i,k}^{\text{UER}}$, $X_{i,k}^{\text{PER}}$, $X_{i,k}^{\text{RR-C}}$, and $X_{i,k}^{\text{RR-M}}$ denote the number of times transition $T_i$ is sampled up to timestep $t = k$ when using uniform experience replay with with-replacement sampling (UER), prioritized experience replay with with-replacement sampling (PER), RR-C, and RR-M, respectively.

## B.2  Bias of RR-C

First, we show that RR-C can be biased when the skipping mechanism in the index buffer (`rr_buffer` in figure 1) is activated.

**Theorem 3.** *There exists a configuration such that $\mathbb{E}\left[X_{i,k}^{RR\text{-}C}\right] \neq \mathbb{E}\left[X_{i,k}^{UER}\right]$ for some $(i, k)$.*

*Proof.* Let $B = R = 1, C = 2$ and $(i, k) = (0, 1)$.

In uniform experience replay, $T_0$ is sampled at $k = 0$ with probability 1 and at $k = 1$ with probability $1/2$, so $\mathbb{E}\left[X_{0,1}^{\text{UER}}\right] = 1 + 1/2 = 3/2$.

In RR-C, at $t = 0$, the index buffer is initialized as either $[0, 1]$ or $[1, 0]$ with probability $1/2$ each.

- (a) If the index buffer is $[0, 1]$, then $T_0$ is sampled at $t = 0$, leaving the index buffer as $[1]$. At $t = 1$, $T_1$ is sampled.

- (b) If the index buffer is $[1, 0]$, then 1 is skipped and $T_0$ is sampled at $t = 0$. Now that the index buffer is empty, the index buffer is regenerated as either $[0, 1]$ or $[1, 0]$ with probability $1/2$ each.

  - (b1) If the regenerated index buffer is $[0, 1]$, then $T_0$ is sampled at $t = 1$.
  - (b2) If the regenerated index buffer is $[1, 0]$, then $T_1$ is sampled at $t = 1$.

(a), (b1), and (b2) occur with probability $1/2$, $1/4$, and $1/4$, respectively. Thus, $\mathbb{E}\left[X_{0,1}^{\text{RR-C}}\right] = 1 \cdot (1/2 + 1/4) + 2 \cdot (1/4) = 5/4 \neq 3/2$. $\square$

However, RR-C is unbiased for sufficiently large $i$ so that the skipping mechanism is never activated.

**Theorem 1.** $\mathbb{E}\left[X_{i,k}^{RR\text{-}C}\right] = \mathbb{E}\left[X_{i,k}^{UER}\right]$ *holds for all $(i, k)$ for sufficiently large $i$.*

*Proof.* First, we note that for timestep $t \geq C$, the replay buffer is always full. To exhaust the whole index buffer of size $C$, we need $\lceil C/B \rceil$ minibatches, i.e. $\lceil C/B \rceil$ timesteps. Thus, there must be a timestep $t^*$ in $C \leq t^* \leq C + \lceil C/B \rceil$ such that the index buffer is regenerated at $t^*$ and there is no skipping thereafter. For

any $i > t^\star$ and any $k \geq i$, all minibatch draws that can include $T_i$ occur at timesteps $t \in [i, \min\{k, i+C-1\}]$ with $t > t^\star$, so the skipping mechanism never affects $X_{i,k}^{\text{RR-C}}$.

In uniform experience replay, as long as the buffer is full, i.e., $i \geq C$, when we sample a transition, the probability of it being transition $T_i$ is $1/C$ as long as it is in the buffer. Thus, for $i \geq C$, $\mathbb{E}\left[X_{i,k}^{\text{UER}}\right] = S_{i,k} \cdot (1/C)$ for all $k \geq i$.

To analyze the case of RR-C, we decompose $S_{i,k}$ as $S_{i,k} = l + mC + n$, where $m$ is a number of times the index buffer is fully exhausted, $l$ is the number of samples taken before the first exhaustion, and $n$ is the number of samples taken after the last exhaustion. We can consider the index buffer is uniformly shuffled when $i$ is sufficiently large, as argued above. Thus, the expected number of times $T_i$ appears in the first $l$ draws is $l/C$, and in the last $n$ draws is $n/C$. Each of the $m$ full exhaustions samples $T_i$ exactly once. Therefore, $\mathbb{E}\left[X_{i,k}^{\text{RR-C}}\right] = l/C + m + n/C = (l + m * C + n)/C = S_{i,k}/C = \mathbb{E}\left[X_{i,k}^{\text{UER}}\right]$. □

Thus, RR-C can be biased only early in training. We have not quantified the bias further, but our numerical simulations in figures 3 to 6 suggest that the bias is negligible in practice.

## B.3   Variance of RR-C

Next, we show that RR-C achieves lower variance in the number of times each transition is sampled compared to uniform experience replay under practical conditions.

**Theorem 2.** $\text{Var}\left[X_{i,k}^{RR\text{-}C}\right] \leq \text{Var}\left[X_{i,k}^{UER}\right]$ *holds for all $(i,k)$ such that $i \leq k$ with sufficiently large $i$. The equality holds only under a few specific conditions.*

*Proof.* As in the proof of unbiasedness of RR-C, we consider the case where $i$ is sufficiently large so that there is no skipping in the index buffer.

In uniform experience replay, whether each sample is $T_i$ is an independent Bernoulli trial with success probability $1/C$. Since we draw $S = B \cdot \min(k - i + 1, C)$ samples from $t = i$ to $t = k$, we obtain $\text{Var}\left[X_{i,k}^{\text{UER}}\right] = S_{i,k} \cdot (1/C) \cdot (1 - 1/C)$.

In RR-C, we again use the decomposition $S_{i,k} = l + mC + n$ from the proof of unbiasedness of RR-C. The variance of $X_{i,k}^{\text{RR-C}}$ comes from the first $l$ samples and the last $n$ samples, as the $m$ full exhaustions always include $T_i$. The event that $T_i$ appears in the first $l$ draws and the event that it appears in the last $n$ draws correspond to different epochs (different independent permutations of the index buffer), so these two Bernoulli variables are independent. Thus, $\text{Var}\left[X_{i,k}^{\text{RR-C}}\right] = (l/C) \cdot (1 - l/C) + (n/C) \cdot (1 - n/C)$.

Now we compare the two variances.

$$\text{Var}\left[X_{i,k}^{\text{UER}}\right] - \text{Var}\left[X_{i,k}^{\text{RR-C}}\right] = S_{i,k} \cdot (1/C) \cdot (1 - 1/C) - (l/C) \cdot (1 - l/C) - (n/C) \cdot (1 - n/C)$$
$$= \frac{l(l-1) + n(n-1) + mC(C-1)}{C^2}$$

Since $0 \leq l < C, 0 \leq m, 0 \leq n < C$, we obtain $\text{Var}\left[X_{i,k}^{\text{RR-C}}\right] \leq \text{Var}\left[X_{i,k}^{\text{UER}}\right]$, where the equality holds only under either of the following conditions: (a) $C = 1$ or (b) $l \in \{0, 1\}, m = 0, n \in \{0, 1\}$. □

The condition (a) is trivial as the buffer can contain only one transition. The condition (b) corresponds to the case where (b1) $i = k$ or (b2) $i + 1 = k$ and the index buffer is regenerated immediately after $t = i$. As we are interested in the variance over multiple timesteps, the condition (b) is not relevant in practice.

Our numerical simulations in figures 3 to 6 support this theoretical result, showing that RR-C achieves lower variance throughout training. While our analysis does not cover early timesteps, the variance for the transitions from early timesteps is also reduced compared to uniform experience replay in our simulations.

### B.4 Bias of RR-M

Similarly to RR-C, we show that RR-M can be biased.

**Theorem 4.** *There exists a configuration such that* $\mathbb{E}\left[X_{i,k}^{RR\text{-}M}\right] \neq \mathbb{E}\left[X_{i,k}^{PER}\right]$ *for some* $(i,k)$.

*Proof.* Let $B = 1, R = 2, C = 3, p_0 = 0.6, p_1 = 0.4, p_2 = 0.0$ and $(i,k) = (0,2)$.

In prioritized experience replay, $T_0$ is sampled at $t = 1, 2$ with probability 0.6 so $\mathbb{E}\left[X_{0,2}^{\text{PER}}\right] = 0.6 + 0.6 = 1.2$.

In RR-M, at $t = 1$, $T_0$ is sampled with probability 0.6 and $T_1$ with 0.4.

- (a) If $T_0$ is sampled at $t = 1$, then the expected and the actual counts are $[0.6, 0.4], [1, 0]$, respectively. At $t = 2$, since the actual count of $T_0$ is greater than the expected count, $T_0$ is masked out and $T_1$ is sampled.

- (b) If $T_1$ is sampled at $t = 1$, then the expected and the actual counts are $[0.6, 0.4], [0, 1]$, respectively. At $t = 2$, since the actual count of $T_1$ is greater than the expected count, $T_1$ is masked out and $T_0$ is sampled.

Thus, $\mathbb{E}\left[X_{0,2}^{\text{RR-M}}\right] = 0.6 \cdot 1 + 0.4 \cdot 1 = 1 \neq 1.2$. $\qquad\square$

Unlike RR-C, this bias can be nonzero even for large $i$, e.g., when $p_i = 0.6, p_{i+1} = 0.4$ and other priorities are zero, the same argument as above applies for all $(i, i+1)$.

Due to the dynamic masking mechanism of RR-M, it is difficult to analyze its bias accurately. Below, we consider a simplified setting where we stop adding a new transition somewhere after $t = i$ so that $T_i$ is forever kept in the buffer. We also assume $B = 1$ for simplicity.

**Lemma 5.** $1 - C < X_{i,k}^{RR\text{-}M} - \mathbb{E}\left[X_{i,k}^{PER}\right] < 1$ *holds for all* $(i,k)$ *under a simplified setting.*

*Proof.* Define $A_i(t)$ and $E_i(t)$ as the actual and expected counts of $T_i$ at timestep $t$ after increment, respectively. Then, we have $X_{i,k}^{\text{RR-M}} = A_i(k)$ and $\mathbb{E}\left[X_{i,k}^{\text{PER}}\right] = E_i(k)$.

By construction of RR-M, $T_i$ is never sampled at $t$ when $A_i(t-1) > E_i(t-1)$, which prevents $A_i(t)$ to be incremented further. Thus, we have $A_i(t) < E_i(t) + 1$ for all $t$, which implies the upper bound, $X_{i,k}^{\text{RR-M}} - \mathbb{E}\left[X_{i,k}^{\text{PER}}\right] < 1$.

Also, $\sum_i A_i(t) = \sum_i E_i(t)$ must hold for any $t$, since both actual and expected counts are incremented by exactly one in total at each timestep. Using this fact and $A_i(t) < E_i(t) + 1$, we have the lower bound $1 - C < X_{i,k}^{\text{RR-M}} - \mathbb{E}\left[X_{i,k}^{\text{PER}}\right]$ $\qquad\square$

Thus, under the simplified setting, the empirical sampling frequency of RR-M converges to that of PER, in the sense of vanishing relative error.

**Theorem 6.** $\lim_{k\to\infty} \frac{X_{i,k}^{RR\text{-}M}}{\mathbb{E}\left[X_{i,k}^{PER}\right]} = 1$ *holds almost surely for all* $i$ *under a simplified setting.*

*Proof.* From lemma 5, we have $|A_i(t) - E_i(t)| < K$, where $K$ is a constant independent of $t$. Dividing both sides by $E_i(t)$ and taking the limit $t \to \infty$, we obtain the result since $E_i(t) \to \infty$ as $t \to \infty$. $\qquad\square$

While we only have this asymptotic result in a simplified setting, our numerical simulations in figures 3 to 6 suggest that the bias is negligible in practice.

### B.5 Variance of RR-M

Using the simplified setting as in the previous subsection, we can also bound the variance of RR-M.

**Lemma 7.** $\mathrm{Var}\left[X_{i,k}^{RR\text{-}M}\right] \leq \frac{1}{4}C^2$ *holds for all* $(i,k)$ *under a simplified setting.*

*Proof.* From theorem 6, we have $X_{i,k}^{\mathrm{RR\text{-}M}} \in \left[1 - C - \mathbb{E}\left[X_{i,k}^{\mathrm{PER}}\right], 1 - \mathbb{E}\left[X_{i,k}^{\mathrm{PER}}\right]\right]$. We apply Popoviciu's inequality on variances to $X_{i,k}^{\mathrm{RR\text{-}M}}$. $\qquad\square$

Note that this variance bound is independent of $k$, meaning that the variance does not grow with the number of samples.

We can now compare the variance of RR-M with that of prioritized experience replay.

**Theorem 8.** $\mathrm{Var}\left[X_{i,k}^{RR\text{-}M}\right] < \mathrm{Var}\left[X_{i,k}^{PER}\right]$ *holds for all* $(i,k)$ *with sufficiently large* $k$ *under a simplified setting.*

*Proof.* It follows from lemma 7 and the fact that the variance of prioritized experience replay grows linearly with the number of samples as $\mathrm{Var}\left[X_{i,k}^{\mathrm{PER}}\right] = (k - i + 1) \cdot p_i \cdot (1 - p_i)$. $\qquad\square$

Our numerical simulations in figures 3 to 6 suggest that the variance of RR-M is lower than that of prioritized experience replay throughout training for more realistic settings.

## C Example trajectories of RR-M

We provide example trajectories of RR-M to illustrate its behavior. We use the simple example from section 3.3, where three transitions with priorities $[1, 0.5, 2]$ are in the buffer, and we sequentially sample 7 transitions. Two trajectories simulated using different random seeds are shown in table 4. Both trajectories sample transitions in different orders, but eventually they sample the first transition twice, the second once, and the third four times, which is the expected behavior of RR-M.

Table 4: Example trajectories of RR-M, where there are three transitions with priorities $[1, 0.5, 2]$ in the buffer and we sequentially sample 7 transitions. Two trajectories simulated using different random seeds are shown.

| iteration | 0 | | | 1 | | | 2 | | | 3 | | | 4 | | | 5 | | | 6 | | |
|---|---|---|---|---|---|---|---|---|---|---|---|---|---|---|---|---|---|---|---|---|---|
| actual counts | 0 | 0 | 0 | 0 | 0 | 1 | 1 | 0 | 1 | 1 | 1 | 1 | 1 | 1 | 2 | 1 | 1 | 3 | 2 | 1 | 3 |
| expected counts | 0.00 | 0.00 | 0.00 | 0.29 | 0.14 | 0.57 | 0.57 | 0.29 | 1.14 | 0.86 | 0.43 | 1.71 | 1.14 | 0.57 | 2.29 | 1.43 | 0.71 | 2.86 | 1.71 | 0.86 | 3.43 |
| mask | F | F | F | F | F | T | T | F | F | T | T | F | F | T | F | F | T | T | T | T | F |
| masked probabilities | 0.29 | 0.14 | 0.57 | 0.67 | 0.33 | 0.00 | 0.00 | 0.20 | 0.80 | 0.00 | 0.00 | 1.00 | 0.33 | 0.00 | 0.67 | 1.00 | 0.00 | 0.00 | 0.00 | 0.00 | 1.00 |
| sampled transition | 2 | | | 0 | | | 1 | | | 2 | | | 2 | | | 0 | | | 2 | | |
| updated actual counts | 0 | 0 | 1 | 1 | 0 | 1 | 1 | 1 | 1 | 1 | 1 | 2 | 1 | 1 | 3 | 2 | 1 | 3 | 2 | 1 | 4 |
| updated expected counts | 0.29 | 0.14 | 0.57 | 0.57 | 0.29 | 1.14 | 0.86 | 0.43 | 1.71 | 1.14 | 0.57 | 2.29 | 1.43 | 0.71 | 2.86 | 1.71 | 0.86 | 3.43 | 2.00 | 1.00 | 4.00 |
| iteration | 0 | | | 1 | | | 2 | | | 3 | | | 4 | | | 5 | | | 6 | | |
| actual counts | 0 | 0 | 0 | 0 | 0 | 1 | 0 | 1 | 1 | 1 | 1 | 1 | 1 | 1 | 2 | 2 | 1 | 2 | 2 | 1 | 3 |
| expected counts | 0.00 | 0.00 | 0.00 | 0.29 | 0.14 | 0.57 | 0.57 | 0.29 | 1.14 | 0.86 | 0.43 | 1.71 | 1.14 | 0.57 | 2.29 | 1.43 | 0.71 | 2.86 | 1.71 | 0.86 | 3.43 |
| mask | F | F | F | F | F | T | F | T | F | T | T | F | F | T | F | T | T | F | T | T | F |
| masked probabilities | 0.29 | 0.14 | 0.57 | 0.67 | 0.33 | 0.00 | 0.33 | 0.00 | 0.67 | 0.00 | 0.00 | 1.00 | 0.33 | 0.00 | 0.67 | 0.00 | 0.00 | 1.00 | 0.00 | 0.00 | 1.00 |
| sampled transition | 2 | | | 1 | | | 0 | | | 2 | | | 0 | | | 2 | | | 2 | | |
| updated actual counts | 0 | 0 | 1 | 0 | 1 | 1 | 1 | 1 | 1 | 1 | 1 | 2 | 2 | 1 | 2 | 2 | 1 | 3 | 2 | 1 | 4 |
| updated expected counts | 0.29 | 0.14 | 0.57 | 0.57 | 0.29 | 1.14 | 0.86 | 0.43 | 1.71 | 1.14 | 0.57 | 2.29 | 1.43 | 0.71 | 2.86 | 1.71 | 0.86 | 3.43 | 2.00 | 1.00 | 4.00 |

## D On the computational efficiency of RR-M

As we discussed in the main text, RR-M requires $O(n)$ time every time it samples a minibatch, where $n$ is the number of transitions in the buffer. We provide a more detailed explanation of the computational cost of RR-M and how it can be implemented efficiently.

## D.1 Computational cost of RR-M

RR-M tracks actual and expected sample counts, updating them every time a minibatch is sampled. Actual counts are incremented only for sampled transitions, requiring $O(b)$ time, where $b$ is the minibatch size. In contrast, expected counts are updated for all transitions in the buffer, so it takes $O(n)$ time. Identifying oversampled transitions involves comparing the actual and expected counts, which also requires $O(n)$ time.

There could be at most $n$ oversampled transitions, and each priority update takes $O(\log n)$ time in a sum tree. Thus, naively masking their priorities with a single sum tree would take $O(n \log n)$ time. This can be avoided by keeping two sum trees, one for the original priorities and the other for the masked priorities, as well as a binary mask that indicates whether each transition is currently oversampled. Masked priorities need updating only for the transitions whose oversampled status has changed. There are at most $b$ transitions that have switched to be oversampled, and there are on average the same number of transitions that have switched in the opposite direction. Therefore, the time complexity of updating masked priorities can be $O(b \log n)$.

In summary, provided the minibatch size is significantly smaller than the buffer size, the time complexity of RR-M is dominated by $O(n)$ operations.

## D.2 Efficient implementation of RR-M

Although updating expected counts and the identification of oversampled transitions each require $O(n)$ time, both tasks can be efficiently parallelized on GPUs. We implement two sum trees, actual and expected sample counts, and a binary mask as GPU arrays using CuPy (Okuta et al., 2017), so we can update them efficiently. Our implementation is included in the supplementary materials.

Although we have not explored this in our experiments, further acceleration could be achieved by introducing some approximation. For example, we could update the expected counts and identify the oversampled transitions only after every $k$ minibatches, where $k$ is a hyperparameter, assuming that the priorities of the transitions will not change significantly within $k$ minibatches. This approach would reduce computational costs by a factor of $k$.

# E Experimental details

## E.1 Additional simulation results

We provide additional results of experience replay simulations with different parameters in figures 4 to 6.

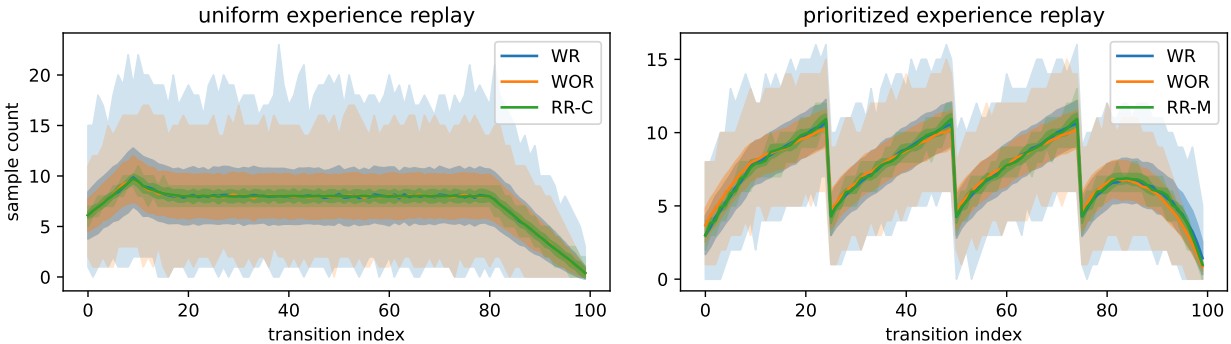

Figure 4: Distributions of sample counts in experience replay simulations with a different set of parameters from figure 3: the minibatch size is 8. Format follows figure 3.

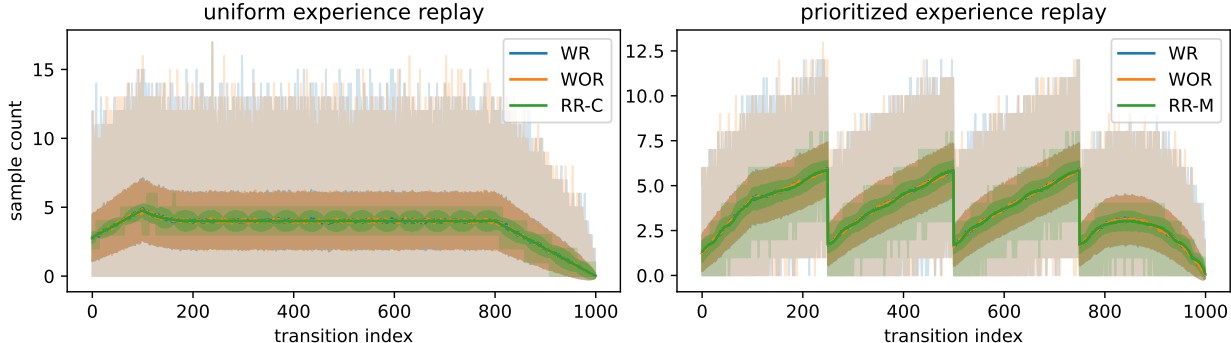

Figure 5: Distributions of sample counts in experience replay simulations with a different set of parameters from figure 3: the total timesteps is 1000, the capacity of the buffer size is 200, the size at which replay starts is 100, and $p_t = (t \bmod 250) + 50$. Format follows figure 3.

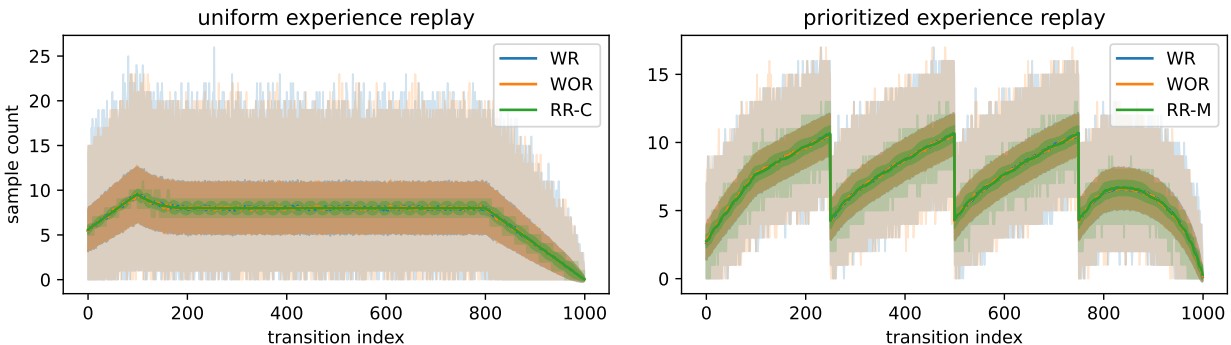

Figure 6: Distributions of sample counts in experience replay simulations with a different set of parameters from figure 3: the total timesteps is 1000, the capacity of the buffer size is 200, the size at which replay starts is 100, the minibatch size is 8, and $p_t = (t \bmod 250) + 50$. Format follows figure 3.

### E.2 Experimental setup

For deep RL experiments with uniform experience replay, we adapt CleanRL's JAX-based implementations of C51 and DQN (Huang et al., 2022):

- C51: `https://github.com/vwxyzjn/cleanrl/blob/1ed80620842b4cdeb1edc07e12825dff18091da9/cleanrl/c51_atari_jax.py`

- DQN: `https://github.com/vwxyzjn/cleanrl/blob/1ed80620842b4cdeb1edc07e12825dff18091da9/cleanrl/dqn_atari_jax.py`

For deep RL experiments with prioritized experience replay, we adapt the official implementation of DDQN+LAP (Fujimoto et al., 2020): `https://github.com/sfujim/LAP-PAL/blob/a649b1b977e979ed0e49d67335d3c79b555e9cfb/discrete/main.py`.

We use the default hyperparameters of the original code unless otherwise noted. Our code for the deep RL experiments is available at `http://github.com/pfnet-research/errr`.

### E.3 Final performance in details

We provide the final performance of C51, DQN, and DDQN+LAP with different sampling methods in the same way as table 1 except that we report the mean ± standard deviations (ddof=1) in tables 5 to 9.

Table 5: Final performance of C51 with with-replacement sampling (WR) and RR-C. Format follows table 1. Mean ± standard deviation (ddof=1) is reported.

|  | C51 WR | C51 RR-C | p-value |
|---|---|---|---|
| Amidar | 347.63±61.80 | **420.61**±52.97 | 1.1e-02* |
| Bowling | **39.45**±8.02 | 33.90±12.23 | 2.5e-01 |
| Frostbite | 3568.36±225.49 | **4073.23**±645.30 | 3.9e-02* |
| KungFuMaster | 21091.90±2181.97 | **21407.00**±2471.17 | 7.7e-01 |
| Riverraid | 11984.34±910.74 | **12776.65**±852.50 | 6.0e-02 |
| BattleZone | 22483.00±1021.12 | **23776.00**±1205.15 | 1.9e-02* |
| DoubleDunk | **-15.04**±0.99 | -16.42±1.43 | 2.2e-02* |
| NameThisGame | 8287.14±358.54 | **8844.65**±325.02 | 1.9e-03** |
| Phoenix | 11920.93±1306.10 | **13749.39**±1416.65 | 7.7e-03** |
| Qbert | 15685.83±742.83 | **16352.80**±771.19 | 6.4e-02 |

Table 6: Final performance of DQN with with-replacement sampling (WR) and RR-C. Format follows table 1. Mean ± standard deviation (ddof=1) is reported.

|  | DQN WR | DQN RR-C | p-value |
|---|---|---|---|
| Amidar | 319.04±29.06 | **341.44**±61.32 | 3.2e-01 |
| Bowling | 42.17±6.48 | **43.97**±9.32 | 6.2e-01 |
| Frostbite | **2351.80**±794.45 | 1897.50±882.38 | 2.4e-01 |
| KungFuMaster | 3674.70±6043.96 | **4291.40**±6823.80 | 8.3e-01 |
| Riverraid | 8346.39±655.53 | **8513.25**±783.58 | 6.1e-01 |
| BattleZone | 22651.00±1178.15 | **24546.00**±1371.04 | 3.9e-03** |
| DoubleDunk | -17.86±5.27 | **-14.69**±5.71 | 2.1e-01 |
| NameThisGame | 6064.81±448.55 | **6919.35**±287.76 | 1.3e-04** |
| Phoenix | 9059.42±1617.43 | **10194.60**±805.68 | 6.8e-02 |
| Qbert | 13191.77±1076.35 | **13859.60**±628.38 | 1.1e-01 |

Table 7: Final performance of C51 with with-replacement sampling (WR) and within-minibatch without-replacement sampling (WOR). Format follows table 1. Mean ± standard deviation (ddof=1) is reported.

|  | C51 WR | C51 WOR | p-value |
|---|---|---|---|
| Amidar | **347.63**±61.80 | 342.91±27.90 | 8.3e-01 |
| Bowling | **39.45**±8.02 | 36.58±6.54 | 3.9e-01 |
| Frostbite | **3568.36**±225.49 | 3487.60±198.76 | 4.1e-01 |
| KungFuMaster | **21091.90**±2181.97 | 19852.80±2957.34 | 3.0e-01 |
| Riverraid | 11984.34±910.74 | **12325.34**±468.77 | 3.1e-01 |
| BattleZone | 22483.00±1021.12 | **22984.00**±1031.61 | 2.9e-01 |
| DoubleDunk | **-15.04**±0.99 | -16.52±0.91 | 2.6e-03** |
| NameThisGame | 8287.14±358.54 | **8466.74**±290.86 | 2.4e-01 |
| Phoenix | **11920.93**±1306.10 | 11587.56±1311.67 | 5.8e-01 |
| Qbert | **15685.83**±742.83 | 15368.27±549.21 | 2.9e-01 |

Table 8: Final performance of DDQN+LAP with with-replacement sampling (WR) and RR-M. Format follows table 1. Mean ± standard deviation (ddof=1) is reported.

|  | DDQN+LAP WR | DDQN+LAP RR-M | p-value |
|---|---|---|---|
| Amidar | **196.94**±24.30 | 195.35±30.06 | 9.0e-01 |
| Bowling | 28.85±5.60 | **29.87**±3.39 | 6.3e-01 |
| Frostbite | 1602.00±204.60 | **1761.34**±167.51 | 7.3e-02 |
| KungFuMaster | 16628.90±1500.14 | **17580.60**±1274.08 | 1.4e-01 |
| Riverraid | 7609.97±283.06 | **7756.11**±120.62 | 1.6e-01 |
| BattleZone | **22897.00**±1570.29 | 20813.00±4352.74 | 1.8e-01 |
| DoubleDunk | -17.49±1.02 | **-17.47**±0.78 | 9.5e-01 |
| NameThisGame | 2589.54±166.60 | **2805.85**±130.60 | 4.9e-03** |
| Phoenix | 4180.98±167.30 | **4342.48**±273.12 | 1.3e-01 |
| Qbert | 4128.27±476.63 | **4266.98**±685.46 | 6.1e-01 |

Table 9: Final performance of DDQN+LAP with stratified sampling (ST) and RR-M+ST. Format follows table 1. Mean ± standard deviation (ddof=1) is reported.

|  | DDQN+LAP ST | DDQN+LAP RR-M ST | p-value |
|---|---|---|---|
| Amidar | 181.64±16.19 | **206.07**±26.32 | 2.5e-02* |
| Bowling | **34.86**±8.33 | 27.57±5.84 | 3.8e-02* |
| Frostbite | 1672.56±214.35 | **1684.80**±211.25 | 9.0e-01 |
| KungFuMaster | 16859.30±1379.02 | **17337.90**±1515.28 | 4.7e-01 |
| Riverraid | 7474.01±260.02 | **7810.06**±321.56 | 2.0e-02* |
| BattleZone | 22099.00±1019.70 | **22584.00**±2232.24 | 5.4e-01 |
| DoubleDunk | -17.34±1.58 | **-17.18**±1.52 | 8.2e-01 |
| NameThisGame | 2635.89±172.90 | **2759.89**±146.11 | 1.0e-01 |
| Phoenix | 4214.82±307.08 | **4399.41**±206.47 | 1.3e-01 |
| Qbert | **4109.27**±412.74 | 4017.55±496.86 | 6.6e-01 |

### E.4 Learning curves

We provide the learning curves of C51, DQN, and DDQN+LAP with different sampling methods in figures 7 to 11.

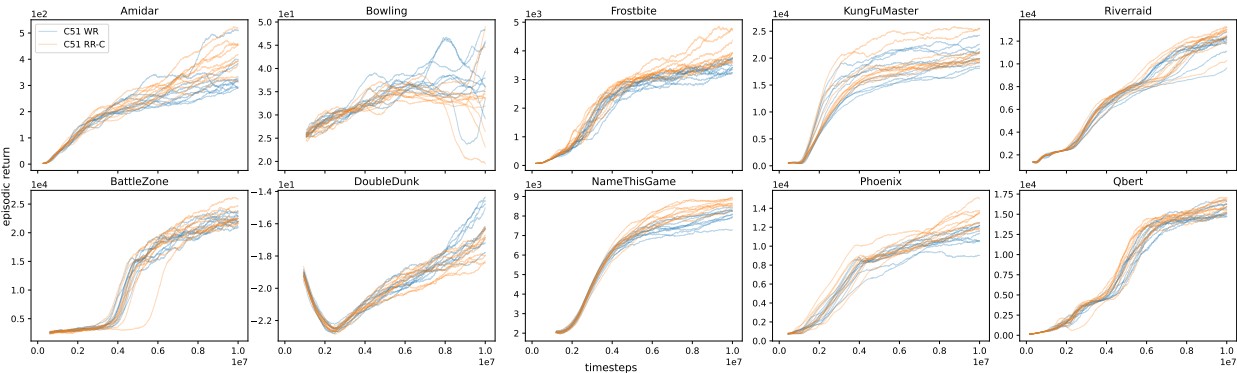

Figure 7: Learning curves of C51 with with-replacement sampling (WR) and RR-C. The 500-episode rolling mean of episodic returns is plotted for each of 10 random seeds.

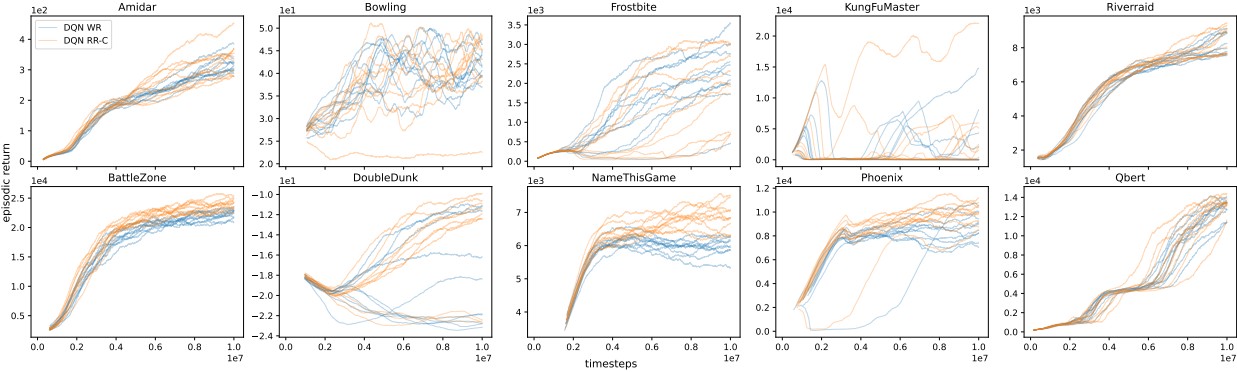

Figure 8: Learning curves of DQN with with-replacement sampling (WR) and RR-C. The 500-episode rolling mean of episodic returns is plotted for each of 10 random seeds.

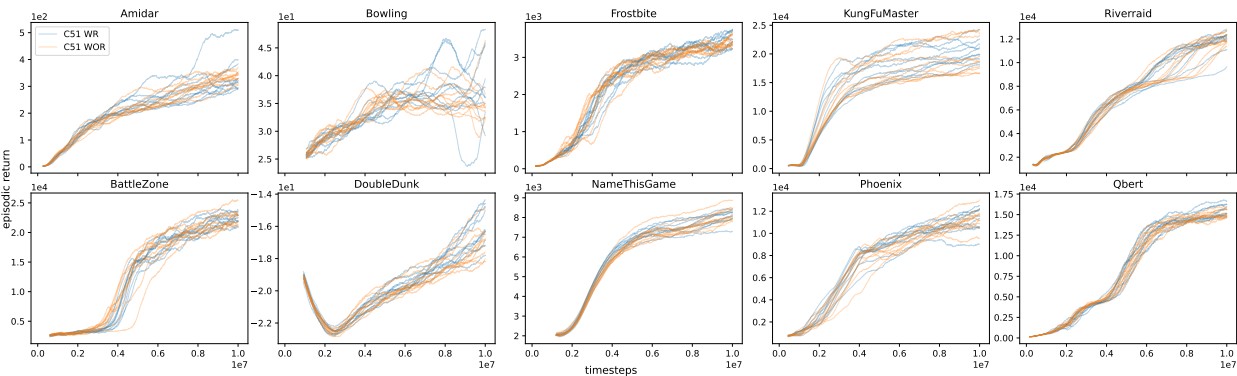

Figure 9: Learning curves of C51 with with-replacement sampling (WR) and within-minibatch without-replacement sampling (WOR). The 500-episode rolling mean of episodic returns is plotted for each of 10 random seeds.

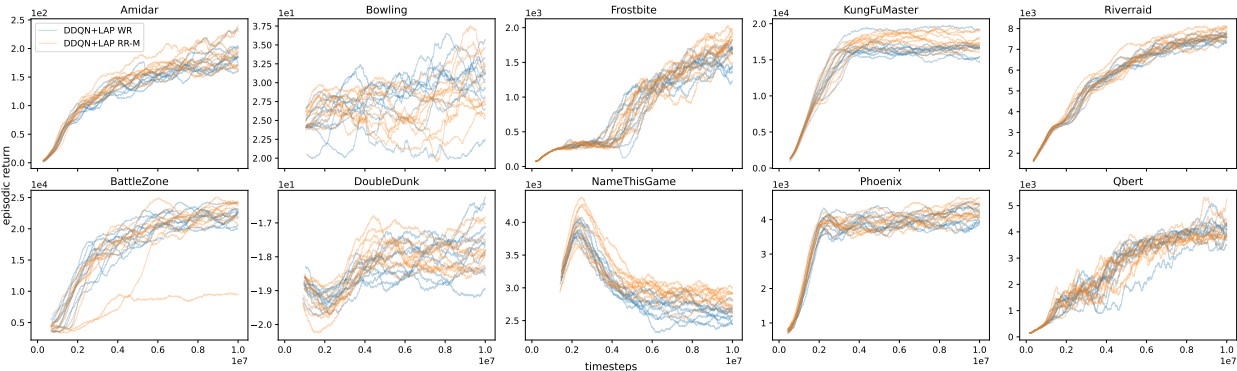

Figure 10: Learning curves of DDQN+LAP with with-replacement sampling (WR) and RR-M. The 500-episode rolling mean of episodic returns is plotted for each of 10 random seeds.

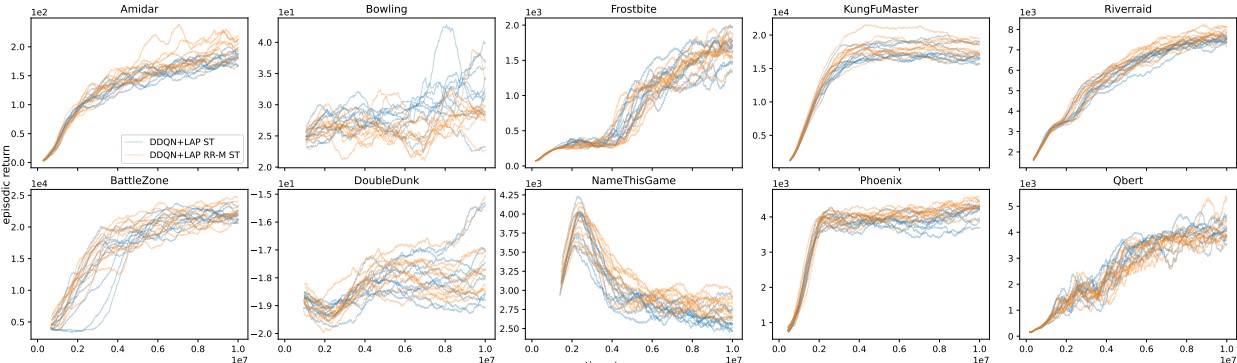

Figure 11: Learning curves of DDQN+LAP with stratified sampling (ST) and RR-M+ST. The 500-episode rolling mean of episodic returns is plotted for each of 10 random seeds.

### E.5 Predicting 57-game median human-normalized scores

We provide the 57-game median human-normalized score predictions of C51, DQN, and DDQN+LAP with different sampling methods following the approach of Aitchison et al. (2023).

We use the random and human scores and the regression coefficients of Atari-10 from Aitchison et al. (2023). We use the last 100-episode mean of episodic returns as the final performance of each run and calculate the human-normalized score for each run using the formula $100 \times \frac{x-r}{h-r}$, where $x$ is the final performance of the run, and $r$ and $h$ are the random and human scores of the game, respectively. Then, we predict the median human-normalized score across 57 games for each run. To estimate the uncertainty of the predictions due to randomness, we use bootstrapping by repeating the following procedure 10,000 times for each algorithm-sampling method pair to obtain a distribution of the median human-normalized score predictions.

1. For each run with a human-normalized score $z$, compute the log-normalized score $\phi(z) = \log_{10}(1 + \max(0, z))$.

2. For each game, we sub-sample 10 log-normalized scores from the 10 runs of different random seeds with replacement and compute their mean $\tilde{\phi}$.

3. We multiply $\tilde{\phi}$ by the regression coefficient of the game, sum the results across games, and apply the inverse transformation $\phi^{-1}(x) = 10^x - 1$ to obtain a prediction of the median human-normalized score.

The top row of figure 12 shows the distributions of the 57-game median human-normalized score predictions for C51, DQN, and DDQN+LAP with different sampling methods. For DQN and DDQN+LAP, RR-C and RR-M show higher predictions than with-replacement sampling. Notably, for C51, RR-C shows lower predictions than with-replacement sampling, which may look inconsistent with the results in table 5. It is also counter-intuitive that minibatch-level without-replacement sampling (WOR) shows much lower predictions, considering that it is expected to perform similarly to with-replacement sampling. This can be attributed to the score of the game DoubleDunk, where WR, WOR, and RR-C achieves mean scores of -15.04, -16.52, and -16.42, respectively. In this game, the gap between the random (-18.55) and human (-16.4) scores is relatively small, so the human-normalized score is sensitive to small score differences that may or may not be due to randomness. To confirm this explanation, we show the prediction results when the regression coefficient for DoubleDunk is set to 0 in the bottom row of figure 12, where C51 behaves similarly to DQN and DDQN+LAP, with RR-based sampling methods showing higher predictions.

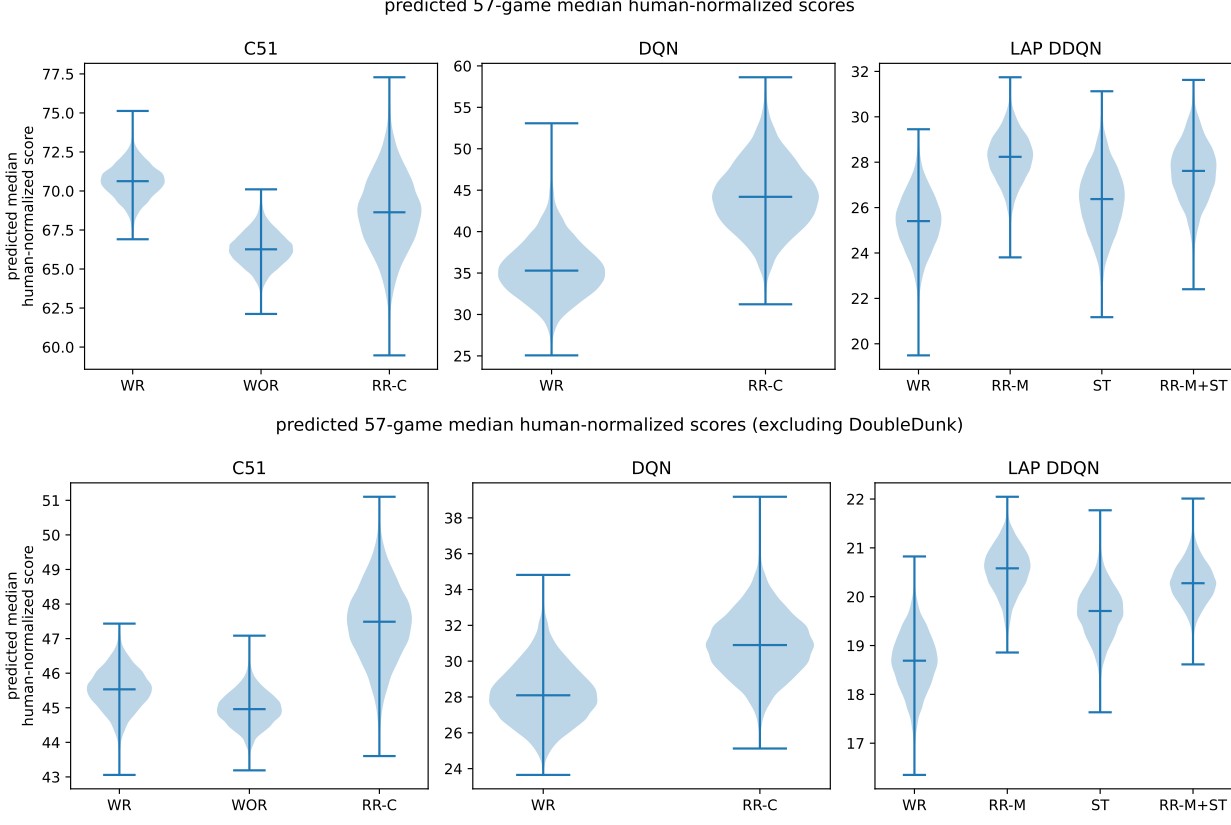

Figure 12: Top row: Predicted 57-game median human-normalized scores of C51, DQN, and DDQN+LAP with different sampling methods based on the regression coefficients of Atari-10 from Aitchison et al. (2023). The distributions of the predictions are obtained by bootstrapping to estimate the uncertainty due to randomness. Bottom row: The same as the top row but subtracting the contribution of the game DoubleDunk by setting its regression coefficient to 0.

