# OpenReview forum: "Experience Replay with Random Reshuffling"
_TMLR — Rejected by TMLR_

### Review · Reviewer_3iq4 · 2025-12-24

**Summary Of Contributions:**

The paper introduces two methods that bring Random Reshuffling to Experience Replay in RL, one for uniform Experience Replay, and the other for Prioritized Experience Replay. Furthermore the authors prove and show through experimental results that using these methods reduces the variance in how often transitions are sampled and used to update the learner, and also that using these methods leads to some small, but statistically significant, performance improvement across some Atari games.

**Audience:**

Yes

**Audience Explanation:**

The work focuses on bringing some useful ideas from Supervised Learning into the world of Reinforcement Learning, while it does not result in a huge change in performance it's still an interesting data point in understanding the relationship between these 2 extremely important settings in machine learning.

**Claims And Evidence:**

No

**Claims Explanation:**

> For uniform experience replay, we propose RR-C, which applies RR to circular buffer indices rather
than transitions themselves, ensuring balanced utilization while maintaining computational efficiency.

Their proposal indeed improves the balance of utilization, i.e. reduces the variance of the frequency of sampling of transitions, while not increasing significantly the computational cost.

> For prioritized experience replay, we develop RR-M, which tracks actual versus expected sample
counts and masks out priorities of oversampled transitions, providing the variance reduction benefits
of RR while respecting priority-based sampling.

RR-M also has the variance reudcing effect while maintaining priority-based sampling, I will add that it seems to be quite more computationally expensive, especially as you increase the replay buffer size, but the authors do not claim that their method is efficient, and so their claim is justified.

> We analyze our methods both theoretically and through simulations, illustrating their bias and
variance properties compared to standard methods.

The authors prove that under weak assumptions the bias introduced by their methods is 0, and that their methods reduce the variance, furthermore they run simulations which show empirical results matching the theory.

> We empirically demonstrate through experiments on Atari benchmarks that both RR-C and RR-M
provide modest performance improvements across different deep RL algorithms: DQN, C51, and
DDQN+LAP.

The authors did not follow the protocol of Aitchison et al., 2023, where one should normalize and then log-normalize the scores of the games, followed by multiplying them by the correct coefficients, as such their results on Atari-10 are not truly representative of the expected performance in the rest of the Atari benchmark.

Furthermore by only using fixed hyperparameters it's unclear whether their methods won't suffer if one must change the size of the experience replay buffer, and whether they're accidentally making the algorithms more brittle to hyperparameter choices.

**Requested Changes:**

I request the authors correct their methodology on using Atari-10, that they add the standard deviation of the performance of the agents in each of the environments to their results table, and that they experiment with more values of the experience replay buffer so we can better understand the robustness of their methods to different hyperparameter values.

---

> ### Author Response · Authors · 2026-02-14
> **Author response to Reviewer 3iq4**
>
> We sincerely thank Reviewer 3iq4 for their careful reading of our paper, the constructive feedback, and for recognizing the value of bridging ideas between supervised learning and reinforcement learning. We address each point below.
>
> ## On the Atari-10 evaluation protocol
>
> We appreciate this important observation. To address it, we have added a new Appendix E.5 in the revised manuscript that reports and discusses predicted 57-game median human-normalized scores following the protocol from Aitchison et al. (2023).
>
> That said, we would like to clarify that we adopted the Atari-10 subset as a relatively small but diverse set of popular environments suitable for empirical evaluation. We do not aim to estimate the 57-game median human-normalized score or claim that our methods' superiority on that metric. We think that evaluating and comparing scores within each game is sufficient to demonstrate the effectiveness of our methods across different environments, which is the main focus of our experiments. We have revised the manuscript to clarify this point.
>
> ## On standard deviation of the performance
>
> Thank you for this suggestion. Standard deviations were already reported in the Appendix tables (Tables 5–9), but we agree they should be easier to find. We have revised the captions of Tables 1 and 2 to direct readers to the corresponding Appendix tables with full standard deviations.
>
> ## On experimenting with different replay buffer sizes
>
> This is a valuable suggestion. While we were unable to run additional large-scale Atari experiments due to computational resource constraints, we believe the following points address the concern about robustness to buffer sizes:
>
> 1. Our theoretical results (Theorems 1, 2 for RR-C; Theorems 6, 8 for RR-M) hold for any buffer capacity C > 1.
> 2. Our simulation experiments (Figures 3–6 in Appendix) cover buffer sizes of 20 and 200 with minibatch sizes of 4 and 8, spanning different ratios of buffer size to minibatch size. In all configurations, RR-based methods consistently reduce sample count variance without noticeable bias.
> 3. Our methods introduce no additional hyperparameters and serve as drop-in replacements for with-replacement sampling at any buffer size.
>
> We acknowledge that varying buffer sizes in the deep RL setting would further strengthen the paper, and we note this as future work.
>
> ## On the alignment between claims and evidence
>
> We note that the reviewer answered "No" to whether claims are supported, raising concerns on the evaluation protocol and replay buffer size variations. We respectfully suggest that the raised concerns pertain to additional analyses that would strengthen the paper rather than to inaccuracies in our existing claims. We believe that our claims are appropriately scoped and supported by the evidence we have provided. We have incorporated the reviewer's suggestions in the revision and are happy to further adjust claims if any specific statement is found to be overstated.

---

### Review · Reviewer_uMot · 2025-12-26

**Summary Of Contributions:**

This paper introduces two methods for extending random reshuffling (RR) to experience replay in reinforcement learning (RL). The core methods are as follows:

1. RR-C (random reshuffling with a circular buffer) for uniform experience replay, which ensures each transition is sampled exactly once per epoch.
2. RR-M (random reshuffling by masking) for prioritized experience replay, which dynamically masks transitions that have been oversampled, ensuring balanced utilization of experiences.

The paper also provides theoretical analysis and empirical evaluations using Atari benchmarks, showing that both RR-C and RR-M offer modest performance improvements in deep RL algorithms like DQN, C51, and DDQN+LAP. The contribution of the paper is an interesting step towards more efficient experience replay, and I commend the authors for exploring this idea.

**Additional Comments:**

The topic is engaging, and I truly appreciate the authors' exploration of adapting RR to experience replay. It’s a thoughtful contribution to the field. However, to meet the high standards of TMLR, I believe the paper needs more substantial empirical results and a clearer connection between the methods' theoretical benefits and real-world impact. I look forward to seeing how these methods can be further refined and tested across a broader range of settings.

**Audience:**

Yes

**Audience Explanation:**

The topic of improving experience replay is certainly relevant to those working in reinforcement learning, particularly in the context of large-scale or complex environments. The idea of applying random reshuffling to experience replay is novel, and the theoretical analysis is solid. However, the modest improvements and the limited experimental validation may not be sufficient to generate significant interest for a broad audience.

**Claims And Evidence:**

No

**Claims Explanation:**

The theoretical analysis is sound, and the authors clearly explain the mechanics of their methods. However, the empirical improvements, while positive, are relatively modest. While the results demonstrate that RR-C and RR-M can reduce the variance in sample counts and lead to some improvements in RL performance, these gains are not large enough to justify a major shift in current practice. The paper presents solid ideas but lacks enough substantial, large-scale improvements to meet the bar expected at TMLR.

Additionally, the experimental setup is limited to Atari-10, which may not fully showcase the potential of the methods in more complex or diverse RL environments. A wider range of environments and algorithms would better demonstrate the broader applicability of the proposed techniques.

**Other weaknesses:**

1. The improvements observed in the experiments are valuable but not substantial enough to make a transformative impact on RL practice. The paper would benefit from demonstrating how these improvements translate into more challenging RL environments.
2. The experiments focus mainly on Atari-10, which limits the generalizability of the results. More diverse environments or real-world RL benchmarks could provide better insights into the applicability of RR-C and RR-M.
3. The paper doesn't clearly discuss the trade-offs between variance reduction and potential computational overhead, especially with RR-M, which could be a concern for large-scale RL tasks.
4. While the methods are easy to implement and computationally efficient in some cases, the RR-M method has computational costs that may not be feasible for certain real-time RL applications.

**Requested Changes:**

1. The experiments should be broadened to include more diverse RL environments and algorithms, as well as more challenging benchmarks beyond Atari-10.
2. A more detailed discussion on the trade-offs between performance improvements and computational costs, especially for RR-M, would strengthen the paper. If computational efficiency is a concern, potential solutions or approximations should be discussed.
3. The authors should clearly discuss the significance of the modest improvements observed. Does RR-C and RR-M provide enough of a boost to justify their use in real-world applications, especially in cases where computational resources are limited?
4. The paper could benefit from more detailed discussion or proof regarding the bias introduced by RR-C and RR-M, especially for smaller buffer sizes or early training stages.

---

> ### Author Response · Authors · 2026-02-14
> **Author response to Reviewer uMot**
>
> We thank Reviewer uMot for the thoughtful review and for acknowledging that the direction of adapting random reshuffling to experience replay is interesting and novel. We address each point below.
>
> ## On the size of improvements
>
> We appreciate the honest assessment and agree that the improvements are modest, which is why we explicitly characterize them as such throughout the paper. We believe our claims are well-calibrated to the evidence: we do not claim transformative gains. We note that TMLR's acceptance criteria focus on whether claims are supported by evidence (https://jmlr.org/tmlr/acceptance-criteria.html). If the reviewer finds any specific claims to be overstated, we would be happy to adjust them.
>
> ## On the diversity of environments and algorithms
>
> We agree that broader experimental validation would further strengthen the paper. While we were unable to run additional experiments due to computational constraints, we believe the combination of a game selection, multiple algorithms, theoretical analysis, and simulations provides sufficient evidence for our carefully scoped claims.
>
> ## On the trade-off between variance reduction and computational cost
>
> We appreciate this point and agree that the complexity of RR-M is a concern. We would like to clarify that we already briefly discuss it Section 3.3, where we acknowledge that RR-M is computationally expensive. Detailed discussion is included in Appendix D, where we also discuss potential approaches to reduce the computational cost of RR-M via parallelization and approximation. Quantifying the exact wall-clock trade-offs would depend on specific hardware and environments; we leave a systematic study of this to future work.
>
> ## On the bias for early stages of training
>
> While we show existence of bias in Appendix for both RR-C and RR-M in the very early stages of training, our simulation results in Figure 3-6 suggest that the bias is not noticeable in practice, even during early training where the theoretical guarantees do not yet apply. We agree that having additional theoretical results that can characterize the bias in the early stages of training would be valuable and we note this as future work.
>
> ## On the alignment between claims and evidence
>
> We note that the reviewer answered "No" to whether claims are supported, citing the modesty of improvements and limited experimental scope. We respectfully suggest that our claims are carefully calibrated to the evidence. Per TMLR's acceptance criteria, any gap between claims and evidence can be addressed by adjusting claims rather than necessarily running more experiments. We believe our current claims are well-supported and are happy to further refine them if specific statements are found to be overstated.

---

### Review · Reviewer_J5Gm · 2026-02-01

**Summary Of Contributions:**

This paper investigates random reshuffling (RR) strategies for experience replay (ER) in reinforcement learning (RL). In supervised learning (SL), RR has consistently demonstrated improved empirical performance compared to replacement based sampling schemes. Motivated by these observations, the authors adapt RR to the RL setting and propose two practical implementations: RR with a circular buffer and RR with masking for prioritized experience replay. This direction is both timely and meaningful, as sampling strategies of this kind remain relatively underexplored in the RL literature.



**Strength**

1. The authors present two novel methods to implement RR with ER. This is different from SL literature as the ER changes dynamically through the learning process.

2. The proposed methods are simple enough to implement in practice.

3. The experimental results are solid. The variance of the proposed methods in the examples matches with the predicted results in the theory. Moreover, The authors provide an improved empirical result with the newly proposed shuffling methods of ER for C51 and DQN.

**Weakness**

1. In contrast to the i.i.d. sampling method in SL, the problem in RL is that the data is usually Markovian.  The authors do not discuss about such important distinctions.

2. Even though the research direction is interesting, the reduction of variance or bias is not new as it has been well-studied in the SL literature (especially in the i.i.d. sampling setting).

3. It is unclear how does random circular reshuffling achieves lower variance than uniform random shuffling with replacement method. How does this relate to existing theory of shuffling methods in the SL literature?

**Additional Comments:**

na

**Audience:**

Yes

**Audience Explanation:**

ER is an important technique in RL and has been widely used in practice. However, the design of effective sampling strategies for ER remains relatively underexplored. Consequently, the findings of this paper may be of significant interest to the RL community.

**Claims And Evidence:**

Yes

**Claims Explanation:**

The authors provided both theoretical and empirical analyses to support their claims. On the empirical side, they examine the variance characteristics of the proposed sampling strategies and evaluate their practical effectiveness by comparing the performance of DQN and C51 on the Atari benchmark suite. On the theoretical side, the authors briefly investigated the bias and variance of the proposed method.

**Requested Changes:**

1. The authors could introduce a brief of sketch of proof or key technique for the proof of Theorem 1 or Theorem 2 in the main manuscript.

2. The authors could defer the python code to the appendix and replace it with a pseudo-code.

3. The authors could additionally include the mean performance curve in the learning plots, for example in Figure 8.

4. Providing a conceptual diagram or visual illustration of the proposed methods would improve clarity and help readers better understand the overall procedure.

---

> ### Author Response · Authors · 2026-02-14
> **Author response to Reviewer J5Gm**
>
> We thank Reviewer J5Gm for the positive and constructive review, for recognizing the timeliness and novelty of our contribution, and for the specific suggestions to improve the paper. We address each point below.
>
> ## On Markovian data
>
> This is an excellent point. We have revised the manuscript to mention the Markovian nature of the data in RL as a key distinction from SL in Section 3.1.
>
> EDIT: We have revised again the manuscript to refine the mention to the Markovian nature.
>
> ## On novelty of variance reduction via RR
>
> We agree that variance and bias reduction through without-replacement sampling is well-studied in the SL literature. Our contribution is not a new theoretical result in the SL setting, but rather the non-trivial adaptation of RR to RL's unique challenges: dynamically changing buffer contents and non-stationary priorities.
>
> ## On why RR-C achieves lower variance
>
> The intuition is the same as for RR in SL. The variance of sample counts for RR in SL is zero within each complete epoch, and we think this intuitively explains why RR-C can achieve lower variance when the buffer is changing slowly enough. A more precise discussion can be found in the theoretical analysis of RR-C in Appendix where we take into account partial epochs and changing buffer contents.
>
> ## On requested changes for refining the presentation
>
> We appreciate these specific and actionable suggestions. We were unable to make these changes during the discussion period, but we would like to incorporate them in the next revision.

---

### Decision · Action_Editor_Fa5x · 2026-04-20

**Recommendation:** Reject

**Audience:**

Yes

**Audience Explanation:**

Reviewers generally agreed that the adaptation of random reshuffling to RL is a timely and interesting direction. Reviewer J5Gm highlighted the practical implementation techniques and interesting experimental results as potential value-adds for the community.

**Claims And Evidence:**

No

**Claims Explanation:**

The paper explores adapting random reshuffling to the experience replay mechanism in reinforcement learning. The goal is to improve optimization efficiency and performance through practical implementation techniques tailored for the RL setting.

Both Reviewers uMot and 3iq4 noted that the performance gains are "modest" at best. Reviewer 3iq4 specifically pointed out that using Atari-10 as a "diverse set" is potentially misleading, as it is intended as a specific proxy protocol for the full Atari-57 suite.

Reviewers also expressed concern over the absence of thorough hyperparameter ablations. Specifically, the sensitivity to replay buffer size and the prohibitive computational cost when combined with prioritized replay were not adequately addressed.